# Towards End-to-End Reinforcement Learning of Dialogue Agents for Information Access

## Abstract

This paper proposes *KB-InfoBot* — a multi-turn dialogue agent which helps users search Knowledge Bases (KBs) without composing complicated queries. Such goal-oriented dialogue agents typically need to interact with an external database to access real-world knowledge. Previous systems achieved this by issuing a symbolic query to the KB to retrieve entries based on their attributes. However, such symbolic operations break the differentiability of the system and prevent end-to-end training of neural dialogue agents. In this paper, we address this limitation by replacing symbolic queries with an induced "soft" posterior distribution over the KB that indicates which entities the user is interested in. Integrating the soft retrieval process with a reinforcement learner leads to higher task success rate and reward in both simulations and against real users. We also present a fully neural end-to-end agent, trained entirely from user feedback, and discuss its application towards personalized dialogue agents.

## 1 Introduction

The design of intelligent assistants which interact with users in natural language ranks high on the agenda of current NLP research. With an increasing focus on the use of statistical and machine learning based approaches (Young et al., 2013), the last few years have seen some truly remarkable conversational agents appear on the market (e.g. Apple Siri, Microsoft Cortana, Google Allo). These agents can perform simple tasks, answer factual questions, and sometimes also aimlessly chit-chat with the user, but they still lag far be-

hind a human assistant in terms of both the variety and complexity of tasks they can perform. In particular, they lack the ability to learn from interactions with a user in order to improve and adapt with time. Recently, Reinforcement Learning (RL) has been explored to leverage user interactions to adapt various dialogue agents designed, respectively, for task completion (Gašić et al., 2013), information access (Wen et al., 2016b), and chitchat (Li et al., 2016a).

We focus on KB-InfoBots, a particular type of dialogue agent that helps users navigate a Knowledge Base (KB) in search of an entity, as illustrated by the example in Figure 1. Such agents must necessarily query databases in order to retrieve the requested information. This is usually done by performing semantic parsing on the input to construct a symbolic query representing the beliefs of the agent about the user goal (see, e.g., Wen et al. (2016b) and Williams and Zweig (2016)). We call such an operation a *Hard-KB* lookup. While natural, this approach has two drawbacks: (1) the retrieved results do not carry any information about uncertainty in semantic parsing, and (2) the retrieval operation is non differentiable, and hence the parser and dialog policy are trained separately. This makes online end-to-end learning from user feedback difficult once the system is deployed.

In this work, we propose a probabilistic framework for computing the posterior distribution of the user target over a knowledge base, which we term a *Soft-KB* lookup. This distribution is constructed from the agent's belief about the attributes of the entity being searched for. The dialogue policy network, which decides the next system action, receives as input this full distribution instead of a handful of retrieved results. We show in our experiments that this framework allows the agent to achieve a higher task success rate in fewer dia-

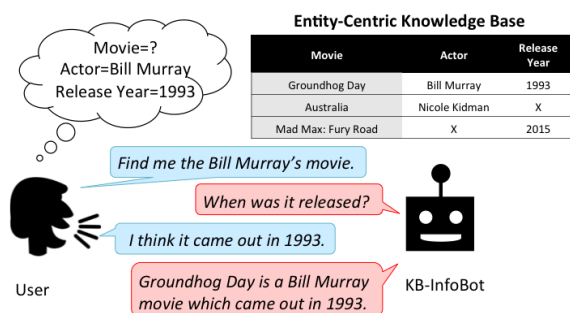

Figure 1: An interaction between a user looking for a movie and the KB-InfoBot. An entity-centric knowledge base is shown above the KB-InfoBot (missing values denoted by **X**).

logue turns. Further, the retrieval process is differentiable, allowing us to construct an end-to-end trainable KB-InfoBot, all of whose components are updated online using RL.

Reinforcement learners typically require an environment to interact with, and hence static dialogue corpora cannot be used for their training. Running experiments on human subjects, on the other hand, is unfortunately too expensive. A common workaround in the dialogue community (Young et al., 2013; Schatzmann et al., 2007b; Scheffler and Young, 2002) is to instead use *user simulators* which mimic the behavior of real users in a consistent manner. For training KB-InfoBot, we adapt the publicly available[1] simulator described in Li et al. (2016b).

Evaluation of dialogue agents has been the subject of much research (Walker et al., 1997; Möller et al., 2006). While the metrics for evaluating an InfoBot are relatively clear — the agent should return the correct entity in a minimum number of turns — the environment for testing it not so much. Unlike previous KB-based QA systems, our focus is on multi-turn interactions, and as such there are no publicly available benchmarks for this problem. We evaluate several versions of KB-InfoBot with the simulator and on real users, and show that the proposed Soft-KB lookup helps the reinforcement learner discover better dialogue policies. Initial experiments on the end-to-end agent also demonstrate its strong learning capability.

## 2 Related Work

Our work is motivated by the neural GenQA (Yin et al., 2016a) and neural enquirer (Yin et al., 2016b) models for querying KBs via natural language in a fully "neuralized" way. However, the

key difference is that these systems assume that users can compose a complicated, compositional natural language query that can uniquely identify the element/answer in the KB. The research task is to parse the query, i.e., turning the natural language query into a sequence of SQL-like operations. Instead we focus on how to query a KB interactively without composing such complicated queries in the first place. Our work is motivated by the observations that (1) users are more used to issuing simple queries of length less than 5 words (Spink et al., 2001); (2) in many cases, it is unreasonable to assume that users can construct compositional queries without prior knowledge of the structure of the KB to be queried.

Also related is the growing body of literature focused on building end-to-end dialogue systems, which combine feature extraction and policy optimization using deep neural networks. Wen et al. (2016b) introduced a modular neural dialogue agent, which uses a Hard-KB lookup, thus breaking the differentiability of the whole system. As a result, training of various components of the dialogue system is performed separately. The intent network and belief trackers are trained using supervised labels specifically collected for them; while the policy network and generation network are trained separately on the system utterances. We retain modularity of the network by keeping the belief trackers separate, but replace the hard lookup with a differentiable one.

Dialogue agents can also interface with the database by augmenting their output action space with predefined API calls (Williams and Zweig, 2016; Zhao and Eskenazi, 2016; Bordes and Weston, 2016). The API calls modify a query hypothesis maintained outside the end-to-end system which is used to retrieve results from this KB. This framework does not deal with uncertainty in language understanding since the query hypothesis can only hold one slot-value at a time. Our approach, on the other hand, directly models the uncertainty to construct the posterior over the KB.

Wu et al. (2015) presented an entropy minimization dialogue management strategy for InfoBots. The agent always asks for the value of the slot with maximum entropy over the remaining entries in the database, which is optimal in the absence of LU errors, and serves as a baseline against our approach. Reinforcement learning neural turing machines (RL-NTM) (Zaremba and Sutskever,

---

[1] https://github.com/MiuLab/UserSimulator

2015) also allow neural controllers to interact with discrete external interfaces. The interface considered in that work is a one-dimensional memory tape, while in our work it is an entity-centric KB.

# 3 Probabilistic KB Lookup

This section describes a probabilistic framework for querying a KB given the agent's beliefs over the fields in the KB.

## 3.1 Entity-Centric Knowledge Base (EC-KB)

A Knowledge Base consists of triples of the form $(h, r, t)$, which denotes that *relation $r$* holds between the *head $h$* and *tail $t$*. We assume that the KB-InfoBot has access to a domain-specific entity-centric knowledge base (EC-KB) (Zwickl-bauer et al., 2013) where all head entities are of a particular type (such as movies or persons), and the relations correspond to attributes of these head entities. Such a KB can be converted to a table format whose rows correspond to the unique head entities, columns correspond to the unique relation types (*slots* henceforth), and some entries may be missing. An example is shown in Figure 1.

## 3.2 Notations and Assumptions

Let $\mathcal{T}$ denote the KB table described above and $\mathcal{T}_{i,j}$ denote the $j$th slot-value of the $i$th entity. $1 \leq i \leq N$ and $1 \leq j \leq M$. We let $V^j$ denote the vocabulary of each slot, i.e. the set of all distinct values in the $j$-th column. We denote missing values from the table with a special token and write $\mathcal{T}_{i,j} = \Psi$. $M_j = \{i : \mathcal{T}_{i,j} = \Psi\}$ denotes the set of entities for which the value of slot $j$ is missing. Note that the user may still know the actual value of $\mathcal{T}_{i,j}$, and we assume this lies in $V^j$. We do not deal with new entities or relations at test time.

We assume a uniform prior $G \sim \mathcal{U}[\{1,...N\}]$ over the rows in the table $\mathcal{T}$, and let binary random variables $\Phi_j \in \{0, 1\}$ indicate whether the user knows the value of slot $j$ or not. The agent maintains $M$ multinomial distributions $p_j^t(v)$ for $v \in V^j$ denoting the probability at turn $t$ that the user constraint for slot $j$ is $v$, given their utterances $U_1^t$ till that turn. The agent also maintains $M$ binomials $q_j^t = \Pr(\Phi_j = 1)$ which denote the probability that the user knows the value of slot $j$.

We assume that column values are independently distributed to each other. This is a strong assumption but it allows us to model the user goal for each slot independently, as opposed to model-

ing the user goal over KB entities directly. Typically $\max_j |V^j| < N$ and hence this assumption reduces the number of parameters in the model.

## 3.3 Soft-KB Lookup

Let $p_{\mathcal{T}}^t(i) = \Pr(G = i|U_1^t)$ be the posterior probability that the user is interested in row $i$ of the table, given the utterances up to turn $t$. We assume all probabilities are conditioned on user inputs $U_1^t$ and drop it from the notation below. From our assumption of independence of slot values, we can write $p_{\mathcal{T}}^t(i) \propto \prod_{j=1}^{M} \Pr(G_j = i)$, where $\Pr(G_j = i)$ denotes the posterior probability of user goal for slot $j$ pointing to $\mathcal{T}_{i,j}$. Marginalizing this over $\Phi_j$ gives:

$$\Pr(G_j = i) = \sum_{\phi=0}^{1} \Pr(G_j = i, \Phi_j = \phi) \quad (1)$$
$$= q_j^t \Pr(G_j = i|\Phi_j = 1) +$$
$$(1 - q_j^t) \Pr(G_j = i|\Phi_j = 0).$$

For $\Phi_j = 0$, the user does not know the value of the slot, and from the prior:

$$\Pr(G_j = i|\Phi_j = 0) = \frac{1}{N}, \quad 1 \leq i \leq N \quad (2)$$

For $\Phi_j = 1$, the user knows the value of slot $j$, but this may be missing from $\mathcal{T}$, and we again have two cases:

$$\Pr(G_j = i|\Phi_j = 1) = \begin{cases} \frac{1}{N_j}, & i \in M_j \\ \frac{p_j^t(v)}{N_j(v)}\left(1 - \frac{|M_j|}{N}\right), & i \notin M_j \end{cases} \quad (3)$$

Here, $N_j(v)$ is the count of value $v$ in slot $j$. Detailed derivation for (3) is provided in Appendix A. Combining (1), (2), and (3) gives us the procedure for computing the posterior over KB entities.

# 4 Towards an End-to-End-KB-InfoBot

We claim that the Soft-KB lookup method has two benefits over the Hard-KB method – (1) it helps the agent discover better dialogue policies by providing it more information from the LU unit, (2) it allows end-to-end training of both dialogue policy and LU in an online setting. In this section we describe several agents to test these claims.

## 4.1 Overview

Figure 2 shows an overview of the components of the KB-InfoBot. At each turn, the agent receives a natural language utterance $u^t$ as input, and selects

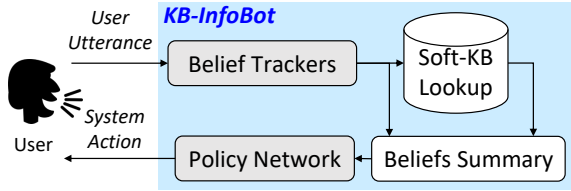

Figure 2: High-level overview of the end-to-end KB-InfoBot. Components with trainable parameters are highlighted in gray.

an action $a^t$ as output. The action space, denoted by $\mathcal{A}$, consists of $M+1$ actions — *request(slot=i)* for $1 \le i \le M$ will ask the user for the value of slot $i$, and *inform(I)* will inform the user with an ordered list of results $I$ from the KB. The dialogue ends once the agent chooses *inform*.

We adopt a modular approach, typical to goal-oriented dialogue systems (see for e.g. (Wen et al., 2016b)), consisting of: a *belief tracker* (LU) module for identifying user intents, extracting associated slots, and tracking the dialogue state (Yao et al., 2014; Hakkani-Tür et al., 2016; Chen et al., 2016; Henderson et al., 2014; Henderson, 2015); an interface with the database to query for relevant results (*Soft-KB* lookup); a *summary* module to summarize the state into a vector; a *dialogue policy* which selects the next system action based on current state (Young et al., 2013). We assume the agent only responds with dialogue acts. A template-based *Natural Language Generator* (NLG) can be easily constructed for converting dialogue acts into natural language.

## 4.2 Belief Trackers

The InfoBot consists of $M$ belief trackers, one for each slot, which get the user input $x^t$ and produce two outputs, $p_j^t$ and $q_j^t$, which we shall collectively call the *belief state*: $p_j^t$ is a multinomial distribution over the slot values $v$, and $q_j^t$ is a scalar probability of the user knowing the value of slot $j$. We describe two versions of the belief tracker.

**Hand-Crafted Tracker:** We first identify mentions of slot-names (such as "actor") or slot-values (such as "Bill Murray") from the user input $u^t$, using token-level keyword search. Let $\{w \in x\}$ denote the set of tokens in a string $x^2$, then for each slot in $1 \le j \le M$ and each value $v \in V^j$, we compute its *matching score* as follows:

$$s_j^t[v] = \frac{|\{w \in u^t\} \cap \{w \in v\}|}{|\{w \in v\}|} \quad (4)$$

---
[2] We use the NLTK tokenizer available at http://www.nltk.org/api/nltk.tokenize.html

A similar score $b_j^t$ is computed for the slot-names. A one-hot vector $req^t \in \{0,1\}^M$ denotes the previously requested slot from the agent, if any. $q_j^t$ is set to 0 if $req^t[j]$ is 1 but $s_j^t[v] = 0 \; \forall v \in V^j$, i.e. the agent requested for a slot but did not receive a valid value in return, else it is set to 1.

Starting from an prior distribution $p_j^0$ (based on the counts of the values in the KB), $p_j^t[v]$ is updated as:

$$p_j^t[v] \propto p_j^{t-1}[v] + C\left(s_j^t[v] + b_j^t + \mathbb{1}(req^t[j] = 1)\right) \quad (5)$$

Here $C$ is a tuning parameter, and the normalization is given by setting the sum over $v$ to 1.

**Neural Belief Tracker:** For the neural tracker the user input $u^t$ is converted to a vector representation $x^t$, using a bag of $n$-grams (with $n = 2$) representation. Each element of $x^t$ is an integer indicating the count of a particular $n$-gram in $u^t$. We let $V^n$ denote the number of unique $n$-grams, hence $x^t \in \mathbb{N}_0^{V^n}$.

Recurrent neural networks have been used for belief tracking (Henderson et al., 2014; Wen et al., 2016b) since the output distribution at turn $t$ depends on all user inputs till that turn. We use a Gated Recurrent Unit (GRU) (Cho et al., 2014) for each tracker, which, starting from $h_j^0 = \mathbf{0}$ computes $h_j^t = \text{GRU}(x^1, \ldots, x^t)$ (see appendix B for details). $h_j^t \in \mathbb{R}^d$ can be interpreted as a summary of what the user has said about slot $j$ till turn $t$. The belief states are computed from this vector as follows:

$$p_j^t = \text{softmax}(W_j^p h_j^t + b_j^p) \quad (6)$$
$$q_j^t = \sigma(W_j^\Phi h_j^t + b_j^\Phi) \quad (7)$$

Here $W_j^p \in \mathbb{R}^{V^j \times d}$, $b_j^p \in \mathbb{R}^{V^j}$, $W_j^\Phi \in \mathbb{R}^d$ and $b_j^\Phi \in \mathbb{R}$, are trainable parameters.

## 4.3 Soft-KB Lookup + Summary

This module uses the Soft-KB lookup described in section 3.3 to compute the posterior $p_\mathcal{T}^t \in \mathbb{R}^N$ over the EC-KB from the belief states $(p_j^t, q_j^t)$. Collectively, outputs of the belief trackers and the soft-KB lookup can be viewed as the current dialogue state internal to the KB-InfoBot. Let $s^t = [p_1^t, p_2^t, \ldots, p_M^t, q_1^t, q_2^t, \ldots, q_M^t, p_\mathcal{T}^t]$ be the vector of size $\sum_j V^j + M + N$ denoting this state. It is possible for the agent to directly use this state vector to select its next action $a^t$. However, the large size of the state vector would lead to a large number of

parameters in the policy network. To improve efficiency we extract summary statistics from the belief states, similar to (Williams and Young, 2005).

Each slot is summarized into an entropy statistic over a distribution $w_j^t$ computed from elements of the KB posterior $p_{\mathcal{T}}^t$ as follows:

$$ w_j^t(v) \propto \sum_{i:\mathcal{T}_{i,j}=v} p_{\mathcal{T}}^t(i) + p_j^0(v) \sum_{i:\mathcal{T}_{i,j}=\Psi} p_{\mathcal{T}}^t(i). \quad (8) $$

Here, $p_j^0$ is a prior distribution over the values of slot $j$, estimated using counts of each value in the KB. The probability mass of $v$ in this distribution is the agent's confidence that the user goal has value $v$ in slot $j$. This two terms in (8) correspond to rows in KB which have value $v$, and rows whose value is unknown (weighted by the prior probability that an unknown might be $v$). Then the summary statistic for slot $j$ is the entropy $H(w_j^t)$. The KB posterior $p_{\mathcal{T}}^t$ is also summarized into an entropy statistic $H(p_{\mathcal{T}}^t)$.

The scalar probabilities $q_j^t$ are passed as is to the dialogue policy, and the final summary vector is $\tilde{s}^t = [H(\tilde{p}_1^t), ..., H(\tilde{p}_M^t), q_1^t, ..., q_M^t, H(p_{\mathcal{T}}^t)]$. Note that this vector has size $2M + 1$.

### 4.4 Dialogue Policy

The dialogue policy's job is to select the next action based on the current summary state $\tilde{s}^t$ and the dialogue history. We present a hand-crafted baseline and a neural policy network.

**Hand-Crafted Policy:** The rule based policy is adapted from (Wu et al., 2015) – it asks for the slot with the minimum entropy, subject to certain constraints listed in Appendix C.

**Neural Policy Network:** For the neural approach, similar to (Williams and Zweig, 2016; Zhao and Eskenazi, 2016), we use an RNN to allow the network to maintain an internal state of dialogue history. Specifically, we use a GRU unit followed by a fully-connected layer and softmax nonlinearity to model the policy $\pi$ over actions in $\mathcal{A}$ ($W^\pi \in \mathbb{R}^{|\mathcal{A}| \times d}$, $b^\pi \in \mathbb{R}^{|\mathcal{A}|}$):

$$ h_\pi^t = \text{GRU}(\tilde{s}^1, ..., \tilde{s}^t) \quad (9) $$

$$ \pi = \text{softmax}(W^\pi h_\pi^t + b^\pi). \quad (10) $$

During training, the agent samples its actions from the policy to encourage exploration. If this action is *inform()*, it must also provide an ordered set of entities indexed by $I = (i_1, i_2, \ldots, i_R)$ in the KB to the user. This is done by sampling $R$ items from the KB-posterior $p_{\mathcal{T}}^t$. This mimics a search engine type setting, where $R$ may be the number of results on the first page.

## 5 Training

Parameters of the neural components (denoted by $\theta$) are trained using the REINFORCE algorithm (Williams, 1992). We assume that the learner has access to a reward signal $r_t$ throughout the course of the dialogue, details of which are in the next section. We can write the expected discounted return of the agent under policy $\pi$ as $J(\theta) = \mathbf{E}_\pi \left[ \sum_{t=0}^{H} \gamma^t r_t \right]$ ($\gamma$ is the discounting factor). When only training the dialogue policy $\pi$ using this signal, updates are given by (details in Appendix D):

$$ \nabla_\theta J(\theta) = \mathbf{E}_\pi \left[ \sum_{k=0}^{H} \nabla_\theta \log \pi_\theta(a^k) \sum_{t=0}^{H} \gamma^t r_t \right], \quad (11) $$

For end-to-end training we need to update both the dialogue policy and the belief trackers using the reinforcement signal, and we can view the retrieval as another policy $\mu_\theta$ (see Appendix D). The updates are given by:

$$ \nabla_\theta J(\theta) = \mathbf{E}_{a \sim \pi, I \sim \mu} \Big[ \big( \nabla_\theta \log \mu_\theta(I) + \sum_{h=0}^{H} \nabla_\theta \log \pi_\theta(a_h) \big) \sum_{k=0}^{H} \gamma^k r_k \Big], \quad (12) $$

In the case of end-to-end learning, we found that for a moderately sized KB, the agent almost always fails if starting from random initialization. In this case, credit assignment is difficult for the agent, since it does not know whether the failure is due to an incorrect sequence of actions or incorrect set of results from the KB. Hence, at the beginning of training we have an *Imitation Learning* (IL) phase where the belief trackers and policy network are trained to mimic the hand-crafted agents. Assume that $\hat{p}_j^t$ and $\hat{q}_j^t$ are the belief states from a rule-based agent, and $\hat{a}^t$ its action at turn $t$. Then the loss function for imitation learning is:

$$ \mathcal{L}(\theta) = \mathbf{E}\big[ D(\hat{p}_j^t || p_j^t(\theta)) + H(\hat{q}_j^t, q_j^t(\theta)) - \log \pi_\theta(\hat{a}^t) \big]. $$

$D(p||q)$ and $H(p,q)$ denote the KL divergence and cross-entropy between $p$ and $q$ respectively.

The expectations are estimated using a mini-batch of dialogues of size $B$. For RL we use RMSProp (Hinton et al., 2012) and for IL we use

vanilla SGD updates to train the parameters $\theta$. For RL we also subtract the mean of the rewards across a batch to reduce the variance in the estimated gradients (Greensmith et al., 2004).

## 6 Experiments and Results

Previous work in KB-based QA has focused on single-turn interactions and is not directly comparable to the present study. Instead we compare different versions of the KB-InfoBot described above to test our claims

### 6.1 KB-InfoBot versions

We have described two belief trackers – (A) Hand-Crafted and (B) Neural, and two dialogue policies – (C) Hand-Crafted and (D) Neural.

**Rule** agents use the hand-crafted belief trackers and hand-crafted policy (A+C). **RL** agents use the hand-crafted belief trackers and the neural policy (A+D). We compare three variants of both sets of agents, which differ only in the inputs to the dialogue policy. The *No-KB* version only takes entropy $H(\hat{p}_j^t)$ of each of the slot distributions. The *Hard-KB* version performs a hard-KB lookup and selects the next action based on the entropy of the slots over retrieved results. This is the same approach as in (Wen et al., 2016b), except that we take entropy instead of summing probabilities. The *Soft-KB* version takes summary statistics of the slots and KB posterior described in Section 4. At the end of the dialogue, all versions inform the user with the top results from the KB posterior $p_{\mathcal{T}}^t$, hence the difference only lies in the policy for action selection. Lastly, the **E2E** agent uses the neural belief tracker and the neural policy (B+D), with a Soft-KB lookup. For the RL agents, we also append $\hat{q}_j^t$ and a one-hot encoding of the previous agent action to the policy network input. Hyperparameter details for the agents are provided in Appendix E.

### 6.2 User Simulator

Training reinforcement learners is challenging because they need an environment to operate in. In the dialogue community it is common to use simulated users for this purpose (Schatzmann et al., 2007a,b; Cuayáhuitl et al., 2005; Asri et al., 2016). In this work we adapt the publicly-available user simulator presented in (Li et al., 2016b) to follow a simple agenda while interacting with the KB-InfoBot, as well as produce natural language

Table 1: Movies-KB statistics for four splits. Refer to Section 3.2 for description of columns.

| KB-split | $N$ | $M$ | $\max_j |V^j|$ | $|M_j|$ |
|---|---|---|---|---|
| Small | 277 | 6 | 17 | 20% |
| Medium | 428 | 6 | 68 | 20% |
| Large | 857 | 6 | 101 | 20% |
| X-Large | 3523 | 6 | 251 | 20% |

utterances[3]. Details about the simulator are included in Appendix F. During training, the simulated user also provides a reward signal at the end of each dialogue. The dialogue is a success if the user target is in top $R = 5$ results returned by the agent; and the reward is computed as $\max(0, 2(1 - (r - 1)/R))$, where $r$ is the actual rank of the target. For a failed dialogue the agent receives a reward of $-1$, and at each turn it receives a reward of $-0.1$ to encourage short sessions[4]. The maximum length of a dialogue is 10 turns beyond which it is deemed a failure.

### 6.3 Movies-KB

We use a movie-centric KB constructed using the IMDBPy[5] package. We constructed four different splits of the dataset, with increasing number of entities, whose statistics are given in Table 1. The original KB was modified to reduce the number of actors and directors in order to make the task more challenging[6]. We randomly remove 20% of the values from the agent's copy of the KB to simulate a scenario where the KB may be incomplete. The user, however, may still know these values.

### 6.4 Simulated User Evaluation

We compare each of the discussed versions along three metrics: the average rewards obtained (R), success rate (S) (where success is defined as providing the user target among top $R$ results), and the average number of turns per dialogue (T). For the RL and E2E agents, during training we fix the model every 100 updates and run 2000 simulations with greedy action selection to evaluate its performance. Then after training we select the model with the highest average reward and run a further 5000 simulations and report the performance in Table 2. For reference we also show the perfor-

---

[3]Simulator & InfoBot code is included in the supplementary material, and will be made public upon acceptance

[4]A turn consists of one user action and one agent action.

[5]http://imdbpy.sourceforge.net/

[6]We restricted the vocabulary to the first few unique values of these slots and replaced all other values with a random value from this set.

Table 2: Performance Comparison. Average (±std error) for 5000 runs after choosing the best model during training. **T:** Average number of turns. **S:** Success rate. **R:** Average reward.

| | Agent | Small KB | | | Medium KB | | | Large KB | | | X-Large KB | | |
|---|---|---|---|---|---|---|---|---|---|---|---|---|---|
| | | T | S | R | T | S | R | T | S | R | T | S | R |
| No KB | Rule | 5.04 | .64 | .26±.02 | 5.05 | .77 | .74±.02 | 4.93 | .78 | .82±.02 | 4.84 | .66 | .43±.02 |
| | RL | 2.65 | .56 | .24±.02 | 3.32 | .76 | .87±.02 | 3.71 | .79 | .94±.02 | 3.64 | .64 | .50±.02 |
| Hard KB | Rule | 5.04 | .64 | .25±.02 | 3.66 | .73 | .75±.02 | 4.27 | .75 | .78±.02 | 4.84 | .65 | .42±.02 |
| | RL | 3.36 | .62 | .35±.02 | **3.07** | .75 | .86±.02 | 3.53 | .79 | .98±.02 | **2.88** | .62 | .53±.02 |
| Soft KB | Rule | **2.12** | .57 | .32±.02 | 3.94 | .76 | .83±.02 | 3.74 | .78 | .93±.02 | 4.51 | .66 | .51±.02 |
| | RL | 2.93 | .63 | .43±.02 | 3.37 | .80 | .98±.02 | 3.79 | .83 | 1.05±.02 | 3.65 | **.68** | **.62±.02** |
| | E2E | 3.13 | **.66** | **.48±.02** | 3.27 | **.83** | **1.10±.02** | **3.51** | **.83** | **1.10±.02** | 3.98 | .65 | .50±.02 |
| *Max* | | *3.44* | *1.0* | *1.64* | *2.96* | *1.0* | *1.78* | *3.26* | *1.0* | *1.73* | *3.97* | *1.0* | *1.37* |

mance of an agent which receives perfect information about the user target without any errors, and selects actions based on the entropy of the slots (*Max*). This can be considered as an upper bound on the performance of any agent (Wu et al., 2015).

In each case the Soft-KB versions achieve the highest average reward, which is the metric all agents optimize. In general, the trade-off between minimizing average turns and maximizing success rate can be controlled by changing the reward signal. Note that, except the E2E version, all versions share the same belief trackers, but by re-asking values of some slots they can have different posteriors $p_{\mathcal{T}}^t$ to inform the results. This shows that having full information about the current state of beliefs over the KB helps the Soft-KB agent discover better policies. Further, reinforcement learning helps discover better policies than the hand-crafted rule-based agents, and we see a higher reward for RL agents compared to Rule ones. This is due to the noisy natural language inputs; with perfect information the rule-based strategy is optimal. Interestingly, the RL-Hard agent has the minimum number of turns in 2 out of the 4 settings, at the cost of a lower success rate and average reward. This agent does not receive any information about the uncertainty in semantic parsing, and it tends to inform as soon as the number of retrieved results becomes small, even if they are incorrect.

Among the Soft-KB agents, we see that E2E>RL>Rule, except for the X-Large KB. For E2E, the action space grows exponentially with the size of the KB, and hence credit assignment gets more difficult. Future work should focus on improving the E2E agent in this setting. The difficulty of a KB-split depends on number of entities it has, as well as the number of unique values for each slot (more unique values make the problem easier). Hence we see that both the "Small" and "X-Large" settings lead to lower reward for the agents, since $\frac{\max_j |V^j|}{N}$ is small for them.

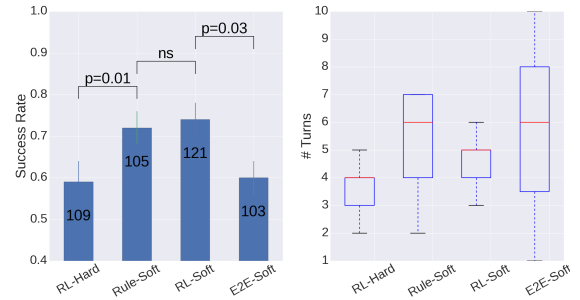

Figure 3: Performance of KB-InfoBot versions when tested against real users. **Left:** Success rate, with the number of test dialogues indicated on each bar, and the p-values from a two-sided permutation test. **Right:** Distribution of the number of turns in each dialogue (differences in mean are significant with $p < 0.01$).

## 6.5 Human Evaluation

We further evaluate the KB-InfoBot versions trained using the simulator against real subjects, recruited from the author's affiliations. In each session, the subject was first presented with a target movie from the "Medium" KB-split along with a subset of its associated slot-values from the KB. To simulate the scenario where end-users may not know slot values correctly, the subjects in our evaluation were presented multiple values for the slots from which they could choose any one while interacting with the agent. Subjects were asked to initiate the conversation by specifying some of these values, and respond to the agent's subsequent requests, all in natural language. We test RL-Hard and the three Soft-KB agents in this study, and in each session one of the agents was picked at random for testing. Figure 3 shows a comparison of these agents in terms of success rate and number of turns, and Figure 4 shows some sample dialogues from the user interactions with RL-Soft.

In comparing Hard-KB versus Soft-KB lookup methods we see that both Rule-Soft and RL-Soft agents achieve a higher success rate than RL-Hard, while E2E-Soft does comparably. They do so in an increased number of average turns, but achieve a

| Turn | Dialogue | Rank | Dialogue | Rank | Dialogue | Rank |
|---|---|---|---|---|---|---|
| 1 | can i get a movie directed by maiellaro *request actor* | 75 | find a movie directed by hemecker *request actor* | 7 | peter greene acted in a family comedy - what was it? *request actor* | 35 |
| 2 | neal *request mpaa_rating* | 2 | i dont know *request mpaa_rating* | 7 | peter *request mpaa_rating* | 28 |
| 3 | not sure about that *request critic_rating* | 2 | i dont know *request critic_rating* | 7 | i don't know that *request critic_rating* | 28 |
| 4 | i don't remember *request genre* | 2 | 7.6 *request critic_rating* | 13 | the critics rated it as 6.5 *inform* | 3 |
| 5 | i think it's a crime movie *inform* | 1 | 7.9 *request critic_rating* | 23 | | |
| 6 | | | 7.7 *inform* | 41 | | |

Figure 4: Sample dialogues between users and the KB-InfoBot (RL-Soft version). Each turn begins with a user utterance followed by the *agent response*. **Rank** denotes the rank of the target movie in the KB-posterior after each turn.

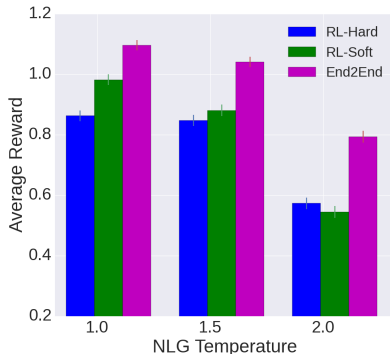

Figure 5: Average rewards against simulator as temperature of softmax in NLG output is increased. Higher temperature leads to more noise in output. Average over 5000 simulations after selecting the best model during training.

higher average reward as well. Between RL-Soft and Rule-Soft agents, the success rate is similar, however the RL agent achieves that rate in a lower number of turns on average. RL-Soft achieves a success rate of 74% on the human evaluation and 80% against the simulated user, indicating minimal overfitting. However, all agents take a higher number of turns against real users as compared to the simulator, due to the noisier inputs.

The E2E gets the highest success rate against the simulator, however, when tested against real users it performs poorly with a lower success rate and a higher number of turns. Since it has more trainable components, this agent is also most prone to overfitting. In particular, the vocabulary of the simulator it is trained against is quite limited ($V^n = 3078$), and hence when real users provided inputs outside this vocabulary, it performed poorly. In the future we plan to fix this issue by employing a better architecture for the LU and Belief Tracker components (such as the one used in Hakkani-Tür et al. (2016)), as well as by pre-training on separate data.

While its generalization performance is poor, the E2E system also exhibits the strongest learning capability. In Figure 5, we compare how different agents perform against the simulator as the temperature of the output softmax in its NLG is increased. A higher temperature means a more uniform output distribution, which leads to generic simulator responses irrelevant to the agent questions. This is a simple way of introducing noise in the utterances. The performance of all agents drops as the temperature is increased, but less so for the E2E agent, which can adapt its belief tracker to the inputs it receives. Such adaptation is key to the personalization of dialogue agents, which motivates us to introduce the E2E agent.

## 7 Conclusions and Discussion

This work is aimed at facilitating the move towards end-to-end trainable dialogue agents for information access. We propose a differentiable probabilistic framework for querying a database given the agent's beliefs over its fields (or slots). We show that such a framework allows the downstream reinforcement learner to discover better dialogue policies by providing it more information. We present an E2E agent for the task, which demonstrates a strong learning capacity in simulations but suffers from overfitting when tested on real users. Given these results, we propose the following deployment strategy that allows a dialogue system to be tailored to specific users via learning from agent-user interactions. The system could start off with an RL-Soft agent (which gives good performance out-of-the-box). As the user interacts with this agent, the collected data can be used to train the E2E agent, which has a strong learning capability. Gradually, as more experience is collected, the system can switch from RL-Soft to the personalized E2E agent. Effective implementation of this, however, requires the E2E agent to learn quickly and this is the research direction we plan to focus on in the future.

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

## A  Posterior Derivation

Here, we present a derivation for equation 3, i.e., the posterior over the KB slot when the user knows the value of that slot. For brevity, we drop $\Phi_j = 0$ from the condition in all probabilities below. For the case when $i \in M_j$, we can write:

$$
\begin{aligned}
\Pr(G_j = i) &= \Pr(G_j \in M_j)\Pr(G_j = i | G_j \in M_j) \\
&= \frac{|M_j|}{N}\frac{1}{|M_j|} = \frac{1}{N} \,,
\end{aligned}
\tag{13}
$$

where we assume all missing values to be equally likely, and estimate the prior probability of the goal being missing from the count of missing values in that slot. For the case when $i = v \notin M_j$:

$$
\begin{aligned}
\Pr(G_j = i) &= \Pr(G_j \notin M_j)\Pr(G_j = i | G_j \notin M_j) \\
&= \left(1 - \frac{|M_j|}{N}\right) \times \frac{p_j^t(v)}{N_j(v)} \,,
\end{aligned}
\tag{14}
$$

where the second term comes from taking the probability mass associated with $v$ in the belief tracker and dividing it equally among all rows with value $v$.

We can also verify that the above distribution is valid: i.e., it sums to 1:

$$
\begin{aligned}
\sum_i \Pr(G_j = i) &= \sum_{i \in M_j} \Pr(G_j = i) + \sum_{i \notin M_j} \Pr(G_j = i) \\
&= \sum_{i \in M_j} \frac{1}{N} + \sum_{i \notin M_j} \left(1 - \frac{|M_j|}{N}\right)\frac{p_j^t(v)}{\#_j v} \\
&= \frac{|M_j|}{N} + \left(1 - \frac{|M_j|}{N}\right)\sum_{i \notin M_j}\frac{p_j^t(v)}{\#_j v} \\
&= \frac{|M_j|}{N} + \left(1 - \frac{|M_j|}{N}\right)\sum_{i \in V^j}\#_j v \frac{p_j^t(v)}{\#_j v} \\
&= \frac{|M_j|}{N} + \left(1 - \frac{|M_j|}{N}\right) \times 1 \\
&= 1 \,.
\end{aligned}
$$

## B  Gated Recurrent Units

A Gated Recurrent Unit (GRU) (Cho et al., 2014) is a recurrent neural network which operates on an input sequence $x_1, \dots, x_t$. Starting from an initial

state $h_0$ (usually set to **0** it iteratively computes the final output $h_t$ as follows:

$$r_t = \sigma(W_r x_t + U_r h_{t-1} + b_r)$$
$$z_t = \sigma(W_z x_t + U_z h_{t-1} + b_z)$$
$$\tilde{h}_t = \tanh(W_h x_t + U_h(r_t \odot h_{t-1}) + b_h)$$
$$h_t = (1 - z_t) \odot h_{t-1} + z_t \odot \tilde{h}_t. \tag{15}$$

Here $\sigma$ denotes the sigmoid nonlinearity, and $\odot$ an element-wise product.

## C  Hand-Crafted Policy

The hand-crafted policy is to ask for the slot $\hat{j} = \arg\min H(\tilde{p}_j^t)$ with the minimum entropy, except if – (i) the KB posterior entropy $H(p_{\mathcal{T}}^t) < \alpha_R$, (ii) $H(\tilde{p}_j^t) < \min(\alpha_T, \beta H(\tilde{p}_j^0))$, (iii) slot $j$ has already been requested $Q$ times. $\alpha_R$, $\alpha_T$, $\beta$, $Q$ are tuned to maximize reward against the simulator.

## D  REINFORCE updates

We assume that the learner has access to a reward signal $r_t$ throughout the course of the dialogue, details of which are in the next section. We can write the expected discounted return of the agent under policy $\pi$ as follows:

$$J(\theta) = \mathbf{E}\left[\sum_{t=0}^{H} \gamma^t r_t\right] \tag{16}$$

Here, the expectation is over all possible trajectories $\tau$ of the dialogue, $\theta$ denotes the trainable parameters of the learner, $H$ is the maximum length of an episode, and $\gamma$ is the discounting factor. We can use the likelihood ratio trick (Glynn, 1990) to write the gradient of the objective as follows:

$$\nabla_\theta J(\theta) = \mathbf{E}\left[\nabla_\theta \log p_\theta(\tau) \sum_{t=0}^{H} \gamma^t r_t\right], \tag{17}$$

where $p_\theta(\tau)$ is the probability of observing a particular trajectory under the current policy. With a Markovian assumption, we can write

$$p_\theta(\tau) = p(s_0) \prod_{k=0}^{H} p(s_{k+1}|s_k, a_k)\pi_\theta(a_k|s_k), \tag{18}$$

where $\theta$ denotes dependence on the neural network parameters. From 17,18 we obtain

$$\nabla_\theta J(\theta) = \mathbf{E}_{a \sim \pi}\left[\sum_{k=0}^{H} \nabla_\theta \log \pi_\theta(a_k) \sum_{t=0}^{H} \gamma^t r_t\right], \tag{19}$$

If we need to train both the policy network and the belief trackers using the reinforcement signal, we can view the KB posterior $p_{\mathcal{T}}^t$ as another policy. During training then, to encourage exploration, when the agent selects the *inform* action we sample $R$ results from the following distribution to return to the user:

$$\mu(I) = p_{\mathcal{T}}^t(i_1) \times \frac{p_{\mathcal{T}}^t(i_2)}{1 - p_{\mathcal{T}}^t(i_1)} \times \cdots. \tag{20}$$

This formulation also leads to a modified version of the episodic REINFORCE update rule (Williams, 1992). Specifically, eq. 18 now becomes,

$$p_\theta(\tau) = \left[p(s_0) \prod_{k=0}^{H} p(s_{k+1}|s_k, a_k)\pi_\theta(a_k|s_k)\right]\mu_\theta(I), \tag{21}$$

Notice the last term $\mu_\theta$ above which is the posterior of a set of results from the KB. From 17,21 we obtain

$$\nabla_\theta J(\theta) = \mathbf{E}_{a \sim \pi, I \sim \mu}\left[\left(\nabla_\theta \log \mu_\theta(I) + \sum_{h=0}^{H} \nabla_\theta \log \pi_\theta(a_h)\right) \sum_{k=0}^{H} \gamma^k r_k\right], \tag{22}$$

## E  Hyperparameters

We use GRU hidden state size of $d = 50$ for the RL agents and $d = 100$ for the E2E, a learning rate of $0.05$ for the imitation learning phase and $0.005$ for the reinforcement learning phase, and minibatch size 128. For the rule agents, hyperparameters were tuned to maximize the average reward of each agent in simulations [TODO:add details]. For the E2E agent, imitation learning was performed for 500 updates, after which the agent switched to reinforcement learning. The input vocabulary is constructed from the NLG vocabulary and bigrams in the KB, and its size is 3078.

## F  User Simulator

At the beginning of each dialogue, the simulated user randomly samples a target entity from the EC-KB and a random combination of *informable slots* for which it knows the value of the target. The remaining slot-values are unknown to the user. The user initiates the dialogue by providing a subset of its informable slots to the agent and requesting for an entity which matches them. In subsequent turns, if the agent requests for the value of a slot,

the user complies by providing it or informs the agent that it does not know that value. If the agent informs results from the KB, the simulator checks whether the target is among them and provides the reward.

We convert dialogue acts from the user into natural language utterances using a separately trained natural language generator (NLG). The NLG is trained in a sequence-to-sequence fashion, using conversations between humans collected by crowd-sourcing. It takes the dialogue actions (DAs) as input, and generates template-like sentences with slot placeholders via an LSTM decoder. Then, a post-processing scan is performed to replace the slot placeholders with their actual values, which is similar to the decoder module in (Wen et al., 2015, 2016a). In the LSTM decoder, we apply beam search, which iteratively considers the top $k$ best sentences up to time step $t$ when generating the token of the time step $t + 1$. For the sake of the trade-off between the speed and performance, we use the beam size of $3$ in the following experiments.

There are several sources of error in user utterances. Any value provided by the user may be corrupted by noise, or substituted completely with an incorrect value of the same type (e.g., "Bill Murray" might become just "Bill" or "Tom Cruise"). The NLG described above is inherently stochastic, and may sometimes generate utterances irrelevant to the agent request. By increasing the temperature of the output softmax in the NLG we can increase the noise in user utterances.

