# Peer review of "Towards End-to-End Reinforcement Learning of Dialogue Agents for Information Access"

_ACL 2017 — decision unknown_

[Official Review · Reviewer 1 · rating 4 · confidence 4]
soundness 3 · originality 4 · clarity 4 · impact 3 · substance 3 · appropriateness 5 · meaningful comparison 5 · presentation format Oral Presentation

This paper presents a dialogue agent where the belief tracker and the dialogue
manager are jointly optimised using the reinforce algorithm. It learns from
interaction with a user simulator. There are two training phases. The first is
an imitation learning phase where the system is initialised using supervising
learning from a rule-based model. Then there is a reinforcement learning phase
where the system has jointly been optimised using the RL objective.

- Strengths: This paper presents a framework where a differentiable access to
the KB is integrated in the joint optimisation. This is the biggest
contribution of the paper. 

- Weaknesses: Firstly, this is not a truly end-to-end system considering the
response generation was handcrafted rather than learnt. Also, their E2E model
actually overfits to the simulator and performs poorly in human evaluation.
This begs the question whether the authors are actually selling the idea of E2E
learning or the soft-KB access. The soft-KB access actually brings consistent
improvement, however the idea of end-to-end learning not so much. The authors
tried to explain the merits of E2E in Figure 5 but I also fail to see the
difference. In addition, the authors didn't motivate the reason for using the
reinforce algorithm which is known to suffer from high variance problem. They
didn't attempt to improve it by using a baseline or perhaps considering the
natural actor-critic algorithm which is known to perform better.

- General Discussion: Apart from the mentioned weaknesses, I think the
experiments are solid and this is generally an acceptable paper. However, if
they crystallised the paper around the idea which actually improves the
performance (the soft KB access) but not the idea of E2E learning the paper
would be better.